# Low-dose aspirin to prevent preeclampsia and growth restriction in nulliparous women identified by uterine artery Doppler as at high risk of preeclampsia: A double blinded randomized placebo-controlled trial

Caroline Diguisto[1,2,3]*, Amelie Le Gouge[4], Marie-Sara Marchand[5], Pascal Megier[6], Yves Ville[7,8], Georges Haddad[9], Norbert Winer[10], Chloé Arthuis[10], Muriel Doret[11], Veronique Houfflin Debarge[12], Anaig Flandrin[13], Hélène Laurichesse Delmas[14], Denis Gallot[14], Pierre Mares[15,16], Christophe Vayssiere[17,18], Loïc Sentilhes[19], Marie-Therese Cheve[20], Anne Paumier[21], Luc Durin[22], Bruno Schaub[23], Veronique Equy[24], Bruno Giraudeau[2,4], Franck Perrotin[1,2], for the Groupe de Recherche en Obstétrique et Gynécologie (GROG)[¶]

1 Pôle de Gynécologie Obstétrique, Médecine Fœtale, Médecine et Biologie de la Reproduction, Centre Olympe de Gouges, CHRU de Tours, Tours, France, 2 Université de Tours, Tours, France, 3 Université de Paris, CRESS, Obstetrical Perinatal and Pediatric Epidemiology Research Team, EPOPé, INSERM, INRAE,F, Paris, France, 4 INSERM CIC1415, CHRU de Tours, Tours, France, 5 Service de Pharmacovigilance CHU Tours, Tours, France, 6 Department of Gynaecology and Obstetrics, Centre Hospitalier Régional d'Orléans, Orleans, France, 7 Centre de Dépistage PRIMA FACIE Université de Paris, Paris, France, 8 Maternité, AP-HP, Hôpital Necker, Paris, France, 9 Cabinet Mosaïque Santé, La Chaussée Saint Victor, France, 10 Department of Obstetrics and Gynecology, University Hospital of Nantes, Nantes, NUN, INRAE, UMR 1280, PhAN, Université de Nantes, Nantes, France, 11 Service de Gynécologie-Obstétrique, HFME, Hospices Civils de Lyon, Lyon, France, 12 Department of Obstetrics, CHU Lille, Univ. Lille, ULR 2694—METRICS: Évaluation des Technologies de Santé et des Pratiques Médicales, Lille, France, 13 Service de Gynécologie-Obstétrique, Hôpital Arnaud de Villeneuve, CHRU de Montpellier, Montpellier, France, 14 Service de Gynécologie-Obstétrique, Hôpital d'Estaing, CHU de Clermont-Ferrand, Maternité Clermont Ferrand, Clermont-Ferrand, France, 15 Département de Gynécologie Obstétrique, Centre Hospitalo-Universitaire Caremeau, Nîmes, France, 16 École de Maïeutique, Université de Montpellier, Site de Nîmes, Nîmes, France, 17 Department of Obstetrics and Gynaecology, Paule de Viguier Hospital, Toulouse University Hospital Center, Toulouse, France, 18 Centre for Epidemiology and Population Health Research, Team SPHERE, Toulouse III University, Toulouse, France, 19 Service de Gynécologie-Obstétrique, Groupe Hospitalier Pellegrin, CHRU de Bordeaux, Talence, France, 20 Service de Gynécologie-Obstétrique, CHR Le Mans, Le Mans, France, 21 Service de Gynécologie-Obstétrique, Polyclinique de l'Atlantique, Saint-Herblain, France, 22 Service de Gynécologie-Obstétrique, Polyclinique du Parc, Caen, France, 23 Service de Gynécologie-Obstétrique, Maison de la Femme, de la Mère et de l'Enfant, CHU Martinique, Fort-de-France, Martinique, France, 24 Service de Gynécologie-Obstétrique, Hôpital Couple Enfant, CHRU de Grenoble, La Tronche, France

¶ Membership of the Groupe de Recherche en Obstétrique et Gynécologie (GROG) is provided in the Acknowledgments.
* carolindiguisto@gmail.com

## Abstract

### Introduction

This trial evaluates whether daily low-dose aspirin initiated before 16 weeks of gestation can reduce preeclampsia and fetal growth restriction in nulliparous women identified by first-trimester uterine artery Dopplers as at high risk of preeclampsia.

**Data Availability Statement:** All relevant data are within the manuscript and its Supporting Information files.

**Funding:** The trial was funded by a government grant "Programme Hospitalier de Recherche Clinique National" PHRCN-2008. The government did not have a role in designing the study, interpreting the results or writing the manuscript.

**Competing interests:** The authors have declared that no competing interests exist.

## Methods

This randomized, blinded, placebo-controlled, parallel-group trial took place in 17 French obstetric departments providing antenatal care. Pregnant nulliparous women aged $\geq$ 18 years with a singleton pregnancy at a gestational age < 16 weeks of gestation with a lowest pulsatility index $\geq$ 1.7 or a bilateral protodiastolic notching for both uterine arteries on an ultrasound performed between 11+0 and 13+6 weeks by a certified sonographer were randomized at a 1:1 ratio to 160 mg of low-dose aspirin or to placebo to be taken daily from inclusion to their 34th week of gestation. The main outcome was preeclampsia or a birthweight $\leq$ 5th percentile. Other outcomes included preeclampsia, severe preeclampsia, preterm preeclampsia, preterm delivery before 34 weeks, mode of delivery, type of anesthesia, birthweight $\leq$ 5th percentile and perinatal death.

## Results

The trial was interrupted due to recruiting difficulties. Between June 2012 and June 2016, 1104 women were randomized, two withdrew consent, and two had terminations of pregnancies. Preeclampsia or a birthweight $\leq$ 5th percentile occurred in 88 (16.0%) women in the low-dose aspirin group and in 79 (14.4%) in the placebo group (proportion difference 1.6 [-2.6; 5.9] p = 0.45). The two groups did not differ significantly for the secondary outcomes.

## Conclusion

Low-dose aspirin was not associated with a lower rate of either preeclampsia or birthweight $\leq$ 5th percentile in women identified by their first-trimester uterine artery Doppler as at high risk of preeclampsia.

## Trial registration

(NCT0172946).

## Introduction

Preeclampsia complicates around 2 to 8% of pregnancies world wild [1, 2]. It can lead to maternal mortality and morbidity, which in turn lead to perinatal morbidity and mortality due to fetal growth restriction (FGR) and medically indicated prematurity [3, 4]. Antiplatelet agents are known to prevent preeclampsia and its consequences when they are administered before 16 weeks of gestation [5, 6]. The difficulty, however, is early identification of pregnancies at high risk of preeclampsia that could benefit from this preventive treatment [7, 8]. Parous women with a history of preeclampsia are candidates, but what is challenging is identifying nulliparous women who should receive this preventive treatment as they have roughly twice the risk to develop preeclampsia when compared to parous women [9, 10].

Preeclampsia and growth restriction are the consequence of abnormal placental implantation and inadequate utero-placental blood flow. Normal placentation comprises trophoblast cell invasion of the decidual and myometrial segments of spiral arteries, which induces reversible changes in the normal arterial wall architecture. Trophoblastic invasion starts from eight weeks' gestation and studies have shown that abnormal uterine artery Doppler as early as the first trimester can identify women at high risk of preeclampsia and fetal growth restriction [11]. The pulsatility index, alone or combined with notching, are the most predictive uterine artery Doppler indices in the first trimester [12].

As women who develop preeclampsia have different uterine artery Doppler patterns on their first-trimester ultrasound [13, 14], we decided to conduct a trial to test the efficacy of low-dose aspirin to reduce the incidence of preeclampsia or growth restriction in nulliparous pregnant women identified at "high-risk" by their first-trimester ultrasound Doppler findings.

## Material and methods

### Study design

We conducted a randomized, blinded, placebo-controlled, parallel-group trial.

### Population

Eligible women were those aged $\geq$ 18 years, with a singleton pregnancy, nulliparous at a gestational age < 16 weeks of gestation with an ultrasound performed between $11^{+0}$ and $13^{+6}$ weeks showing a lowest pulsatility index $\geq$ 1.7 for both uterine arteries or bilateral protodiastolic notching [15]. Women carrying a fetus with severe congenital anomalies (with or without termination of pregnancy) diagnosed on this first-trimester ultrasound, women receiving anticoagulant treatments or with any contraindications to antiplatelet treatment, and those with coagulopathies, lupus, or antiphospholipid syndrome were not included.

### Setting

Recruitment took place in 16 French maternity units including tertiary university hospitals, general hospitals, and private obstetrics departments, and one private imaging center.

### Screening for participants

As the inclusion criteria involved sonographic parameters, sonographers (physicians or midwives) screened the women. All sonographers had validated an online certificate (Collège Français d'Échographie Fœtale) for first-trimester uterine artery Doppler examinations ([13] https://www.epp-echofoetale.fr). Sonographers were invited to an information meeting held in each participating unit before the trial began to review the aims and the protocol of the study. Eligible women were screened and informed of the study during their first trimester ultrasound. All eligible women were seen in a specific consultation with a physician or midwife in the participating unit. This provider explained the aims and scope of the study, checked inclusion and exclusion criteria, and obtained written consent from the women who met all inclusion and no exclusion criteria and were willing to participate.

### Randomization and blinding

Women were randomly allocated in a 1:1 ratio to 160 mg of low-dose aspirin daily or to a placebo. A secure computer-generated, online, centralized web-based system managed the randomization (with a fixed block size of four units, in a sequence generated by a statistician from INSERM CIC 1415 not involved in patient recruitment) and concealment processes. Women, clinicians, and outcome assessors were blinded to allocation. The dataset was unblinded for analysis once the data collection was finalized. In case of a serious adverse event which would require unblinding, this was possible at any time of the study; however this has never been necessary.

### Intervention group

After randomization, hospital pharmacists dispensed to women in the experimental group a sufficient quantity of low-dose aspirin to be taken daily during their evening meal from the

day of inclusion until they reached their 34<sup>th</sup> week of gestation. The purchased product was in the form of 288-g sachets of DLlysine acetylsalicylate powder, corresponding to 160 mg of low-dose aspirin (Kardegic®), to be dissolved in water for administration. They were asked to store it in a dry place at a temperature less than 25˚ Celsius.

### Control group

Women randomized to the control group received placebo powder (purchased from Bertin Pharma), in similar-looking sachets as the intervention group, to be stored and taken according to the same protocol.

### Follow-up

The women received a notebook, which they had to return to the investigating center after childbirth, to indicate any possible side effects, together with a card stating that they were participating in the study, with the contact details required in case of a medical emergency for which an unblinding procedure was absolutely necessary. Follow-up was identical in the two groups and women's pregnancies were monitored and managed by their physician or midwife according to French guidelines. After childbirth, women were asked to bring back all the remaining treatments and packaging to the pharmacists of their center to assess adherence to the treatment.

### Outcomes

The main outcome was a composite of preeclampsia or newborn with a birthweight $\leq$ 5th percentile. Preeclampsia was defined by hypertension in pregnancy (systolic blood pressure $\geq$ 140 mmHg or diastolic pressure $\geq$ 90 mmHg, measured twice at least 4 hours apart), and proteinuria greater than 300 mg/24 hour after 20 weeks or in the post-partum period [16, 17]. To determine birthweights $\leq$ 5th percentile, we used the EPOPé growth curves, adjusted for the newborn's gestational age and sex and the mother's parity, height, and weight [18].

The secondary outcomes were preeclampsia, severe preeclampsia, defined as women with preeclampsia and any of the following: systolic blood pressure $\geq$ 160 mmHg or diastolic blood pressure $\geq$ 110 mmHg, proteinuria $\geq$ 5 g/day, oliguria, HELLP syndrome, eclampsia, acute pulmonary edema, placental abruption, or stillbirth. The rate of preterm preeclampsia (<37 weeks' gestation) was added as a secondary outcome after the trial's registration at clinical-trials.gov to be able to compare results with those of the ASPRE trial [19]. Secondary outcomes also included preterm delivery before 34 weeks, mode of delivery (vaginal or caesarean) and the type of anaesthesia for delivery. Secondary outcomes for infants included a birthweight $\leq$ 5th percentile or perinatal death, defined as a stillbirth from 22 weeks through a neonatal death in the first 7 days. Any kind of bleeding was also studied as a secondary outcome. Serious adverse events, defined as any event, reaction or unexpected adverse reaction that results in death, is life threatening or requires hospitalization were also reported and coded according to MedDra dictionary (version 19.1) (https://admin.meddra.org/sites/default/files/guidance/file/94911910_termselptc_r4_12_sep2016.pdf.)

### Sample size

According to an unpublished study conducted in our unit, the presence of first trimester bilateral notching and/or a high pulsatility index among nulliparas is predictive of the occurrence of pre-eclampsia with a sensitivity of 75% and a specificity of 67% and predictive of growth restriction with a sensitivity of 55% and a specificity of 67%. Based on this performance we hypothesized that low-dose aspirin could reduce the primary outcome, expected to be 21.2%

in the control group (protocol in S3 File), by 15% [20–22]. To show a relative reduction from 21.2% to 18.0% in the occurrence of preeclampsia or birthweight $\leq$ 5th percentile with a power of 80% and a two-tailed type I error of 5%, we required 2415 women in each group. As we expected that some pregnancies would end in termination, and we planned four intermediate analyses (every 1000 inclusions) to counter the unknown precise expected rates of outcome in the control group, we planned to include 2486 women in each group. The study protocol (S3 File) describes the sample size calculation in fuller detail.

## Analysis

A statistical analysis plan was finalized before the dataset was frozen. Because the trial was stopped prematurely, no interim analysis was performed. Women were analysed according to their randomization group (intention to treat). Baseline characteristics were reported per group with numbers and percentages for categorical variables and with medians and interquartile ranges for continuous variables. For the primary outcome, missing data were managed with simple imputation by assuming that women with missing data for the primary outcome had preeclampsia or that their infant had a birthweight $< 5^{th}$ percentile. A complete-case analysis was also conducted. Rates were then compared with the Chi-square test. The between-group difference in proportions was estimated as well as its 95% confidence interval (CI) (Wald method). Results were also presented as crude odd ratios (ORs) with their 95% CIs. No imputation was performed for secondary outcomes, which were also compared with Chi-square tests. The between-group difference and odds ratios were also presented for secondary outcomes. Statistical analyses were performed with SAS version 9.4 and R version 3.3.1 software.

## Ethics

The study protocol (S3 File) and patient information documents were approved by the competent French authorities: Agence Nationale de Sécurité du Médicament et des produits de santé (number of approval A120316-72 on the 07/05/2012) and Comité de Protection des Personnes (number 2012-R8; 27/03/2012). The study protocol was registered at ClinicalTrials.gov (NCT01729468) and in the European EudraCT database (2011-003536-30).

## Results

Between June 2012 and June 2016, 1104 women were randomized. The trial was stopped after 4 years due to recruiting difficulties resulting in a failure to conduct any of the four initially planned intermediate analyses. Because two women withdrew their consent and two terminations of pregnancy occurred after randomization, 1100 women were finally analysed: 550 in the intervention (low-dose aspirin) and 550 in the control (placebo) group (Fig 1).

The mean age of the women were 28.3 ± 4.9 years and 28.7 ± 4.7 years in the aspirin and the placebo group, respectively. Eight percent of women were of white ethnicity, S1 Table. The median gestational age at delivery were 39.7 [38.5–40.6] and 39.7 [38.5–40.8] in the aspirin and the placebo group, respectively. The number (rate) of cases of preeclampsia or infants with a birthweight $\leq$ 5th percentile was 88 (16.0%) in the low-dose aspirin group versus 79 (14.4%) in the placebo group (proportion difference 1.6 [-2.6; 5.9], p = 0.45 OR 1.14 95%CI (0.82–1.58)), S2 Table. These findings were supported by our complete-case analysis, which showed 76 (14.1%) and 72 (13.3%) with preeclampsia or infants with birthweight $\leq$ 5th percentile in the low-dose aspirin and the placebo groups, respectively ($P$ = 0.68). The two groups did not differ for secondary outcomes. Also in each group, during treatment, around 25% of the women reported bleeding (epistaxis or gingival bleeding (80%) and metrorrhagia (20%))

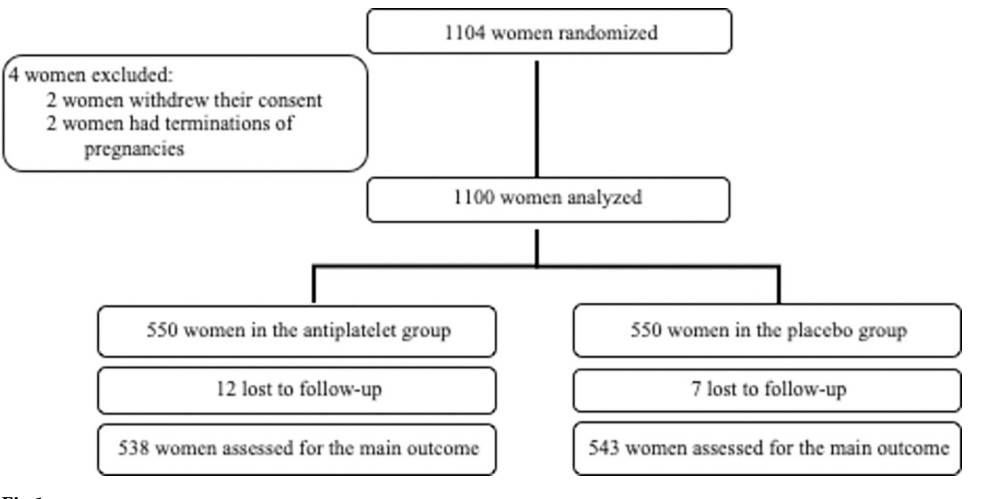

**Fig 1.**

as an adverse effect. Overall, 112 (20%) and 120 (22%) women, respectively, experienced at least one serious adverse event (S3 Table). In all, around 25% of the women in each group stopped their treatment before their 34[th] week.

## Discussion

In this trial conducted among women identified at high risk of preeclampsia by their first-trimester uterine artery Doppler examination, the rate of preeclampsia or birthweight<5[th] percentile in women who received low-dose aspirin before 16 weeks gestation was not statistically different from that of women receiving placebo.

This trial was planned before the results from the ASPRE trial which also evaluated an intervention among women at high risk of preeclampsia [19]. They identified women by using an algorithm including demographic, clinical, biological, and Doppler findings, which produced a population very different from ours, with 30% of the women parous and 10% with a history of preeclampsia. In the ASPRE trial, approximately forty percent of the primary outcomes occurred in parous women. Being limited to nulliparas is the main strength of our trial, as parous women differ substantially in terms of risk assessment for preeclampsia and are not the women for whom the decision about preventive treatment is challenging. Other strengths include that all sonographers were certified for uterine artery Dopplers and also that all women were administered their treatment before 16 weeks of gestation which is the only way the preventive treatment can be effective.

Nevertheless, our trial has limitations, including that we were not able to reach the number of inclusions initially planned, although we extended our inclusion period by one year. This may be explained partly by an inaccurate initial assessment of feasibility and the lower than expected number of eligible women. Recruitment may also have been difficult as healthy nulliparous women may not have understood how their participation could improve outcomes for them or the baby as, by definition, nulliparas have never experienced adverse pregnancy outcomes. Another limitation was the lower than expected rate for the main outcome in the control group (14% of events versus 21% expected). Our hypothesis and sample size were built on numbers of outcomes which were not specific from the French population. A French study conducted when the trial was designed found that preeclampsia affected only 1.7% of nulliparous women whereas we used an expected rate of 6% [10]. It is likely that the lower than

expected rate for the main outcome may be explained, in part, by this erroneous assessment of the rate of preeclampsia in nulliparous women in France [9]. Thus, we can speculate that if an intermediate analysis would have been conducted, it is likely that the trial would have been stopped for futility. One in four woman interrupted their treatment before their 34[th] week of gestation which is concordant with compliance findings from metanalysis evaluating the effect of aspirin to prevent preeclampsia in the pregnant population [5]. The reported rates of women who were fully compliant in the studies included in the metanalysis varied between 60 and 90%. Detailed compliance was not assessed precisely or regularly through the follow up of the pregnancy so we are unable to explain these adherence rates. Also, the screening took place in 17 participating units and we were not able to assess how many first-trimester ultrasound examinations of nulliparous women were conducted by sonographers during the inclusion period. We are therefore unable to estimate how many women were screened. Finally, we suspect some of the adverse events may have been under-reported. Indeed, some of the adverse events were also secondary outcomes (metrorrhagia and preterm delivery) and there are some discrepancies between the numbers of outcomes and adverse events which is why we are cautious in the interpretation our adverse event data.

The observed rate of preterm preeclampsia in our control group was half that in the ASPRE trial control group, which means that their algorithm was more successful in identifying women at risk of preeclampsia. We found another study also focusing on the evaluation of first trimester uterine Doppler among a population of nulliparas who were not particularly at high risk [23]. They found a rate of preeclampsia in their population (4.9%) which was close to what we observed in our study thereby confirming that we were unable to identify a high-risk population. Bujold et al.'s meta-analysis which evaluated acetylsalicylic acid for the prevention of preeclampsia and intra-uterine growth restriction in women with abnormal uterine artery doppler found that this treatment, when initiated early, could reduce the incidence of preeclampsia and in particular severe preeclampsia [24]. It turns out that most of the observed cases of term preeclampsia and was mild which may explain partly the lack effect of the treatment in our trial.

Algorithms that aim to identify women at high risk of preeclampsia by clinical, biological, and ultrasound parameters have been published and have sparked enthusiasm, but the cost effectiveness of their implementation at a population level is still debated [25–28]. The trial was designed in 2008 and at the time, none of those algorithms were recommended which is why we selected our population differently. Until now, those algorithms are still not recommended by national guidelines even if recommended by the International Federation of Gynecology and Obstetrics, the International Society of ultrasound in obstetrics and Gynecology and the International Society for the Study of Hypertension in Pregnancy [29, 30]. Many studies were published during the elapsed time between de design of the trial (2008) and the publication (2022). This delay was mainly due to motivation issues and organisation difficulties. Inclusion criteria to select women at high risk of preeclampsia which seemed appropriate at the time are now obsolete. Because women in most high-income countries are advised to undergo a first-trimester ultrasound, screening for preeclampsia through clinical and Doppler findings would not require a major increase in cost [31]. Accordingly, a combination of clinical and Doppler findings should be used in further studies to evaluate interventions to reduce preeclampsia.

## Conclusion

In our study, low-dose aspirin was not associated with a lower rate than placebo of either preeclampsia or FGR in women identified as at high risk of preeclampsia during a first-trimester

uterine artery Doppler examination. We interpret our results cautiously because of the lack of power related to insufficient number of patients recruited.

## Supporting information

**S1 Table. Characteristics of the trial participants.** Results are numbers and percentages unless stated otherwise.
(DOCX)

**S2 Table. Outcomes according to randomization group.**
(DOCX)

**S3 Table. Serious adverse events.**
(DOCX)

**S1 File. Supporting information CONSORT checklist.**
(DOC)

**S2 File. Supporting information dataset.**
(XLSX)

**S3 File. Supporting information protocol.**
(DOC)

## Acknowledgments

We thank all the women who agreed to participate in the study. We would like to thank Anne Rebion for her analysis, Hélène Bourgoin for her help with the pharmacy. We also thank the research assistants who helped to conduct this trial Yoann Desvignes and Catherine Fermont and the participating members of the "Groupe de Recherche en Obstétrique et Gynécologie (GROG)", Thomas Schmitz, Elie Azria, Catherine Deneux-tharaux, Anne Ego, Francois Goffi-net, Cyril Huissoud, Gilles Kayem, Bruno Langer, Camille Le Ray, Olivier Morel, Marie-Vic-toire Senat and Damien Subtil.

## Author Contributions

**Conceptualization:** Amelie Le Gouge, Marie-Sara Marchand, Yves Ville, Bruno Giraudeau, Franck Perrotin.

**Data curation:** Amelie Le Gouge, Yves Ville, Georges Haddad, Bruno Giraudeau, Franck Perrotin.

**Formal analysis:** Amelie Le Gouge, Chloé Arthuis, Muriel Doret, Franck Perrotin.

**Funding acquisition:** Amelie Le Gouge, Norbert Winer, Franck Perrotin.

**Investigation:** Caroline Diguisto, Pascal Megier, Veronique Houfflin Debarge, Anaig Flan-drin, Hélène Laurichesse Delmas, Denis Gallot, Pierre Mares, Christophe Vayssiere, Marie-Therese Cheve, Anne Paumier, Luc Durin, Bruno Schaub, Veronique Equy, Bruno Girau-deau, Franck Perrotin.

**Methodology:** Chloé Arthuis, Muriel Doret, Bruno Giraudeau, Franck Perrotin.

**Project administration:** Georges Haddad, Norbert Winer, Veronique Houfflin Debarge, Franck Perrotin.

**Resources:** Yves Ville, Georges Haddad, Norbert Winer, Bruno Giraudeau, Franck Perrotin.

**Software:** Pascal Megier, Chloé Arthuis, Veronique Houfflin Debarge, Franck Perrotin.

**Supervision:** Marie-Sara Marchand, Yves Ville, Norbert Winer, Chloé Arthuis, Muriel Doret, Franck Perrotin.

**Validation:** Caroline Diguisto, Marie-Sara Marchand, Pascal Megier, Chloé Arthuis.

**Visualization:** Caroline Diguisto, Muriel Doret.

**Writing – original draft:** Caroline Diguisto, Amelie Le Gouge, Georges Haddad, Norbert Winer, Veronique Houfflin Debarge, Anaig Flandrin, Hélène Laurichesse Delmas, Denis Gallot, Pierre Mares, Christophe Vayssiere, Loïc Sentilhes, Marie-Therese Cheve, Anne Paumier, Luc Durin, Bruno Schaub, Veronique Equy, Bruno Giraudeau.

**Writing – review & editing:** Caroline Diguisto, Amelie Le Gouge, Pascal Megier, Georges Haddad, Veronique Houfflin Debarge, Anaig Flandrin, Hélène Laurichesse Delmas, Denis Gallot, Pierre Mares, Christophe Vayssiere, Loïc Sentilhes, Marie-Therese Cheve, Anne Paumier, Luc Durin, Bruno Schaub, Veronique Equy, Bruno Giraudeau.

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
