## [Decision Letter · Decision Letter 0]

1 Jun 2022

PONE-D-22-07025Low-dose aspirin to prevent preeclampsia and growth restriction in nulliparous women identified by uterine artery Doppler as at high risk of preeclampsia: a double blinded randomized placebo-controlled trialPLOS ONE

Dear Dr. Diguisto,

Thank you for submitting your manuscript to PLOS ONE. After careful consideration, we feel that it has merit but does not fully meet PLOS ONE’s publication criteria as it currently stands. Therefore, we invite you to submit a revised version of the manuscript that addresses the points raised during the review process.

ACADEMIC EDITOR:

As highlighted by Reviewer 3, although this trial was interrupted prematurely, the results of this study are interesting. However, this manuscript needs major modifications. Therefore, responses to reviewers' comments should be done carefully, especially Reviewer 2 comments.

We look forward to receiving your revised manuscript.

Kind regards,

Patrick Rozenberg, MD

Academic Editor

PLOS ONE

Journal Requirements:

2. Thank you for stating the following in the Financial Section of your manuscript:

“The trial was funded by a government grant “Programme Hospitalier de Recherche Clinique National” PHRCN-2008.”

We note that you have provided additional information within the Funding Section that is not currently declared in your Funding Statement. Please note that funding information should not appear in the Funding section or other areas of your manuscript. We will only publish funding information present in the Funding Statement section of the online submission form.

“The trial was funded by a government grant “Programme Hospitalier de Recherche Clinique National” PHRCN-2008. The government did not have a role in designing the study, interpreting the results or writing the manuscript”

“NO authors have competing interests”

5. Please note that in order to use the direct billing option the corresponding author must be affiliated with the chosen institute. Please either amend your manuscript to change the affiliation or corresponding author, or email us at plosone@plos.org with a request to remove this option.

6. One of the noted authors is a group or consortium “for the Groupe de Recherche en Obstétrique et Gynécologie (GROG)”. In addition to naming the author group, please list the individual authors and affiliations within this group in the acknowledgments section of your manuscript. Please also indicate clearly a lead author for this group along with a contact email address.

Additional Editor Comments::

As highlighted by Reviewer 3, although this trial was interrupted prematurely, the results of this study are interesting. However, this manuscript needs major modifications. Therefore, responses to reviewers' comments should be done carefully, especially Reviewer 2 comments

Reviewers' comments:

Reviewer's Responses to Questions

**Comments to the Author**

1. Is the manuscript technically sound, and do the data support the conclusions?

Reviewer #1: Yes

Reviewer #2: Yes

Reviewer #3: Yes

2. Has the statistical analysis been performed appropriately and rigorously? 

Reviewer #1: Yes

Reviewer #2: Yes

Reviewer #3: Yes

3. Have the authors made all data underlying the findings in their manuscript fully available?

Reviewer #1: No

Reviewer #2: Yes

Reviewer #3: Yes

4. Is the manuscript presented in an intelligible fashion and written in standard English?

Reviewer #1: Yes

Reviewer #2: Yes

Reviewer #3: Yes

5. Review Comments to the Author

Reviewer #1: A randomized controlled clinical trial was conducted which aimed to evaluate whether daily low-dose aspirin reduces preeclampsia and growth restriction in nulliparous women at high risk for the condition. The proportion of women who developed preeclampsia was not significantly different in the two randomization arms.

Minor revisions:

1- Line 190: Provide a measure of dispersion for age.

2- Table 1: It is customary to report means and standard deviations for normally distributed variables and median, first and third quartiles for non-normally distributed variables.

3- Table 2: Provide a p-value comparing the association of anesthesia and arm. Perhaps this p-value is 0.94. If so, move the p-value to the line labeled “Anesthesia (n=539 vs 537).”

4- Table 3: Provide p-values comparing the proportions with adverse events.

5- Indicate if adverse events were collected according to a standardized method such as the Common Terminology Criteria for Adverse Events (CTCAE).

Reviewer #2: Low-dose aspirin to prevent preeclampsia and growth restriction in nulliparous women identified by uterine artery Doppler as at high risk of preeclampsia: a double blinded randomized placebo-controlled trial; by Diguisto C et al.

This study addresses a very interesting question: the usefulness of low-dose aspirin (LDA) to prevent preeclampsia and growth restriction in high risk nulliparous women identified by early uterine artery Doppler. The article is well-written and easy to read.

I have some major comments:

In the introduction section the rational of the study in insufficiently detailed:

- Line 82-86, the authors discuss the use of algorithms involving biological, maternal and ultrasound parameters to select high risk women. This is off-topic in the introduction and the authors should mention the articles they are relying on to carry out the study. In addition, in their rational they refer to articles published after the start of the trial, which is illogical (ref 9,11,12)), and they use a reference in which uterine doppler was performed in the second trimester (ref 11)

- Amazingly, I did not find any reference in the introduction section supporting the use of early uterine Doppler alone to select high risk nulliparous women. This is essential for the calculation of the number of patients to be included in the study

- Instead of citing general articles concerning the rate of PE (line 74, ref 1 and 2) the authors are lucky to have French publications on the subject, with a risk of PE in nulliparous women of 1.7% (Subtil et Al, ERASME trial Part 1, BJOG. 2003;110:475-484), and in general population of 0.7% (Goffinet et al. BJOG. 2001;108:510-8.)

- In addition, do the authors have any idea on the performance of early uterine Doppler to select high risk nulliparous women in a french cohort (pilot study for instance)

In the Screening for participants section (line 101) the authors should mention how many women were approached before randomization, the reasons of refusals, and put all these informations in the flow-chart.

In sample size section, I do not understand how the authors calculated the number of patients to recruit. They must provide the articles they rely on because the two references cited by the authors (ref 18 and 19) do not address prediction of SGA < 5th centile. The authors should discuss and may be use the findings of Pilalis et al ( Acta Obstetrica et Gynecologica. 2007; 86:530-34.) where these authors found the same composite outcome of 14%. This article might have helped the author to calculate the number needed to recruit.

In Analysis section, the authors should present the data of an RCT as RR and absolute risk reduction. In addition, they should mention when they did the unblinding compared to the analysis of complications

In the results section:

Table 2, the authors must separate primary outcome from secondary outcomes, add two columns ( one for RR and one for ARR)

They must add a specific par for side effects. In addition, Table 3 is not clear: serious events are different from those of Table 2?, for instance, rates of preterm delivery are different ?; What are the maternal hypertensive complications? Why the numbers are not similar for perinatal death? I encourage the authors to detail serious events.

In the discussion section:

Line 271, add after was not “statistically”…

You must discuss in first why you have such difference of primary outcome (21.2% expected and 14.4% observed), almost 32%.

In the conclusion section, you must add that your results are uncertain because of lack of power related to insufficient number of patients recruited in the study.

Reviewer #3: This is a randomized, blinded, placebo-controlled, parallel-group trial assessing whether daily low-dose aspirin initiated before 16 weeks of gestation can reduce preeclampsia and growth restriction in nulliparous women identified by first-trimester uterine artery Dopplers as at high risk of preeclampsia. Aspirin was administered at the appropriated doe of 160 mg/day. Preeclampsia or a birthweight ≤ 5th percentile occurred in 88 (16.0%) women in the low-dose aspirin group and in 79 (14.4%) in the placebo group (proportion difference 1.6 [-2.6 ; 5.9] p=0.45). The two groups did not differ significantly for the secondary outcomes.

The manuscript by Diguisto et al. is well written. The methodology is clear and clearly explained. Although, the trial was interrupted after including 44% of the expected number of patients (1104 of the 2486 planned), the results of this study are interesting and need to be published.

In the introduction, please mention the prevalence of preeclampsia (PE) in France in the general population and in nulliparous women. This will allow to show that the patients of the placebo group are at higher risk of PE.

The result section is very limited. Please describe the results in this section. Gestational age at delivery should be added.

In the discussion, please present the results of the previous studies evaluating first trimester uterine Doppler (such The Great Obstetrical Syndrome Study) and show how the outcomes of the placebo group are comparable to these studies. Please also discuss the previous studies assessing the effect of low dose aspirin in patients first trimester uterine Doppler and show how the outcomes of the placebo (Acetylsalicylic acid for the prevention of preeclampsia and intra-uterine growth restriction in women with abnormal uterine artery Doppler: a systematic review and meta-analysis; Usefulness of aspirin therapy in high-risk pregnant women with abnormal uterine artery Doppler ultrasound at 14-16 weeks pregnancy: randomized controlled clinical trial).

Please also discuss that 2/3 of preeclampsia cases occurred after 37WG. This is normal but may contribute to the results of the trial.

Line 121 : the sentence is not clear. Did the patients receive Kardegic sachets or a reconditioned for of aspirin? If patients received Kardegic sachets please explain how the patients and the investigators were blinded to the treatment?

Lines 301-303: The sentence is not clear. Please who was more successful in identifying women at risk.

6. PLOS authors have the option to publish the peer review history of their article (what does this mean?). If published, this will include your full peer review and any attached files.

Reviewer #1: No

Reviewer #2: No

Reviewer #3: No

---

## [Author Response · Author response to Decision Letter 0]

15 Jul 2022

The Editor-in-Chief, Plosone

 July, 2022

Dear Editor,

We would be grateful if you would consider our revised manuscript PONE-D-22-07025-R1 “Low-dose aspirin to prevent preeclampsia and growth restriction in nulliparous women identified by uterine artery Doppler as at high risk of preeclampsia: a double blinded randomized placebo-controlled trial “which we hope is now suitable for publication in Plosone. You will find below our responses to each of the editor’s and reviewers’ comments and suggestions. We have changed the manuscript as requested by the editor and reviewers, when appropriate. 

The authors have declared that no competing interests exist.

Caroline Diguisto, MD, PhD, on behalf of all authors

Pôle de gynécologie obstétrique, médecine fœtale, médecine et biologie de la reproduction, centre Olympe de Gouges, CHRU de Tours, 37 044 Tours, France Tel: +33 2 47 47 47 47 

Email carolinediguisto@gmail.com

Thank you for stating the following in the Financial Section of your manuscript:

“The trial was funded by a government grant “Programme Hospitalier de Recherche Clinique National” PHRCN-2008.”

We note that you have provided additional information within the Funding Section that is not currently declared in your Funding Statement. Please note that funding information should not appear in the Funding section or other areas of your manuscript. We will only publish funding information present in the Funding Statement section of the online submission form.

“The trial was funded by a government grant “Programme Hospitalier de Recherche Clinique National” PHRCN-2008. The government did not have a role in designing the study, interpreting the results or writing the manuscript”

The funding statement is accurate, it is fine as it is.

Thank you for stating the following in your Competing Interests section: 

“NO authors have competing interests”

We have added this statement in the cover letter.

We have changed our data availability statement and now provide, as requested a minimal dataset.

5. Please note that in order to use the direct billing option the corresponding author must be affiliated with the chosen institute. Please either amend your manuscript to change the affiliation or corresponding author, or email us at plosone@plos.org with a request to remove this option.

6. One of the noted authors is a group or consortium “for the Groupe de Recherche en Obstétrique et Gynécologie (GROG)”. In addition to naming the author group, please list the individual authors and affiliations within this group in the acknowledgments section of your manuscript. Please also indicate clearly a lead author for this group along with a contact email address.

Participating members of the “Groupe de Recherche en Obstétrique et Gynécologie (GROG)” were Thomas Schmitz, Elie Azria, Catherine Deneux-tharaux, Anne Ego, Francois Goffinet, Cyril Huissoud, Gilles Kayem, Bruno Langer, Camille Le Ray, Olivier Morel, Marie-Victoire Senat and Damien Subtil. Their name has been added in the acknowledgments section. There is no lead author for this group.

Additional Editor Comments:

As highlighted by Reviewer 3, although this trial was interrupted prematurely, the results of this study are interesting. However, this manuscript needs major modifications. Therefore, responses to reviewers' comments should be done carefully, especially Reviewer 2 comments

Comments to the Author

1. Is the manuscript technically sound, and do the data support the conclusions?

Reviewer #1: Yes

Reviewer #2: Yes

Reviewer #3: Yes 

2. Has the statistical analysis been performed appropriately and rigorously? 

Reviewer #1: Yes

Reviewer #2: Yes

Reviewer #3: Yes

3. Have the authors made all data underlying the findings in their manuscript fully available?

Reviewer #1: No

Reviewer #2: Yes

Reviewer #3: Yes

4. Is the manuscript presented in an intelligible fashion and written in standard English?

Reviewer #1: Yes

Reviewer #2: Yes

Reviewer #3: Yes

Reviewer #1: A randomized controlled clinical trial was conducted which aimed to evaluate whether daily low-dose aspirin reduces preeclampsia and growth restriction in nulliparous women at high risk for the condition. The proportion of women who developed preeclampsia was not significantly different in the two randomization arms.

Minor revisions:

1- Line 190: Provide a measure of dispersion for age.

We have added this information and presented the mean age and the standard deviation for both groups “The mean age of the women were 28.3 ± 4.9 years and 28.7 ± 4.7 years in the aspirin and the placebo group, respectively.”

2- Table 1: It is customary to report means and standard deviations for normally distributed variables and median, first and third quartiles for non-normally distributed variables.

We have changed table 1, it turns out only the variable “age” has a normal distribution. It is now presented as a mean and a standard deviation 

3- Table 2: Provide a p-value comparing the association of anesthesia and arm. Perhaps this p-value is 0.94. If so, move the p-value to the line labeled “Anesthesia (n=539 vs 537).”

Indeed 0.94 concerns the anesthesia variable. We have moved 0.94 to the appropriate line

4- Table 3: Provide p-values comparing the proportions with adverse events.

We have compared the rates of patients with at least one serious adverse event and added this information in table 3 (p=0.55). We would rather not compare individually each event as we prefer the strategy of a composite outcome structure, also called the basket which has been proposed to capture the unexpected side effect in clinical trials (Tugwell et al. J Clin Epidemiol 2005;58:785-790). We provide p values only for those adverse events that are secondary outcomes.

5- Indicate if adverse events were collected according to a standardized method such as the Common Terminology Criteria for Adverse Events (CTCAE).

The adverse events were not collected according to a standardized method but were coded according to MedDra dictionary (version 19.1) . This was added in the methods section.

Reviewer #2: Low-dose aspirin to prevent preeclampsia and growth restriction in nulliparous women identified by uterine artery Doppler as at high risk of preeclampsia: a double blinded randomized placebo-controlled trial; by Diguisto C et al.

This study addresses a very interesting question: the usefulness of low-dose aspirin (LDA) to prevent preeclampsia and growth restriction in high risk nulliparous women identified by early uterine artery Doppler. The article is well-written and easy to read.

I have some major comments:

In the introduction section the rational of the study in insufficiently detailed:

- Line 82-86, the authors discuss the use of algorithms involving biological, maternal and ultrasound parameters to select high risk women. This is off-topic in the introduction and the authors should mention the articles they are relying on to carry out the study. 

We have changed the introduction section. The introduction section is now appropriate to the perspective of when the trial was designed. We removed the text which concerns biological parameters. 

In addition, in their rational they refer to articles published after the start of the trial, which is illogical (ref 9,11,12)), and they use a reference in which uterine doppler was performed in the second trimester (ref 11)

We have changed the references and have a reference that refers to uterine artery doppler in the first trimester.

- Amazingly, I did not find any reference in the introduction section supporting the use of early uterine Doppler alone to select high risk nulliparous women. This is essential for the calculation of the number of patients to be included in the study

We have added some references which support the use of early uterine doppler: reference 11 and 12. 

- Instead of citing general articles concerning the rate of PE (line 74, ref 1 and 2) the authors are lucky to have French publications on the subject, with a risk of PE in nulliparous women of 1.7% (Subtil et Al, ERASME trial Part 1, BJOG. 2003;110:475-484), and in general population of 0.7% (Goffinet et al. BJOG. 2001;108:510-8.)

We have added the suggested references in the introduction. They are also mentioned in the discussion section as the numbers we have used to build our hypothesis are slightly different. This difference may explain in part the lower than expected numbers for the main outcome.

- In addition, do the authors have any idea on the performance of early uterine Doppler to select high risk nulliparous women in a french cohort (pilot study for instance)

According to an unpublished study conducted in our unit, which we used to calculate the size of our sample, the presence of bilateral notching and/or a high pulsatility index is predictive of the occurrence of pre-eclampsia among nulliparas, and this with a sensitivity of 75% and a specificity of 67%. According to the same study, the presence of bilateral notching and/or a high pulsatility index is predictive of the occurrence of growth restriction with a sensitivity of 55% and a specificity of 67%, also among nulliparas. Based on those performances to predict preeclampsia and growth restriction, the sample size was calculated as explained in the protocol which we have attached as supplementary file. Unfortunately this study has not been published. We changed the “sample size” section to better explain this.

In the Screening for participants section (line 101) the authors should mention how many women were approached before randomization, the reasons of refusals, and put all these informations in the flow-chart.

Unfortunately, because it is really difficult to maintain correctly a screening log, especially when physicians are from the private sector, the number of women who were approached was unavailable. This is already mentioned in the discussion section “Finally the screening took place in 17 participating units and we were not able to assess how many first-trimester ultrasound examinations of nulliparous women were conducted by sonographers during the inclusion period. We are therefore unable to estimate how many women were screened. “ 

In sample size section, I do not understand how the authors calculated the number of patients to recruit. They must provide the articles they rely on because the two references cited by the authors (ref 18 and 19) do not address prediction of SGA < 5th centile. The authors should discuss and may be use the findings of Pilalis et al ( Acta Obstetrica et Gynecologica. 2007; 86:530-34.) where these authors found the same composite outcome of 14%. This article might have helped the author to calculate the number needed to recruit.

We used a pilot study conducted in our unit to calculate the number of patients to recruit. The performance to predict preeclampsia and growth restriction is explained above and has been used to calculate the size of the sample. The detailed calculations for this sample size are in the protocol, and we have made the sample size section more explicit.

In Analysis section, the authors should present the data of an RCT as RR and absolute risk reduction. 

We agree and in agreement with the CONSORT Statement have added two columns in table 2 to add this information.

In addition, they should mention when they did the unblinding compared to the analysis of complications

We have added this information in the methods section “The dataset was unblinded for analysis once the data collection was finalized. In case of a serious adverse event which would require unblinding, this was possible at any time of the study however this has never been necessary.”

In the results section:

Table 2, the authors must separate primary outcome from secondary outcomes, add two columns ( one for RR and one for ARR)

We have added two columns in table 2 to add this information 

They must add a specific par for side effects. In addition, Table 3 is not clear: serious events are different from those of Table 2?, for instance, rates of preterm delivery are different ?; What are the maternal hypertensive complications? Why the numbers are not similar for perinatal death? I encourage the authors to detail serious events.

Hypertensive disorders were defined as any gestational hypertension or preeclampsia in pregnancy. Some adverse events such as metrorrhagia and preterm birth were also secondary outcomes. Those were more frequently observed as outcomes than as adverse events. This may be due to some under-reporting of adverse events by investigators which has not been. For this reason we are cautious in the interpretation of our data on adverse events. This point has been added in our discussion section. 

In the discussion section:Line 271, add after was not “statistically”…

We have added “statistically”

You must discuss in first why you have such difference of primary outcome (21.2% expected and 14.4% observed), almost 32%.

It is likely that the lower than expected rate for the main outcome may be explained, in part, by this erroneous assessment of the rate of preeclampsia in nulliparous women in France. Our hypothesis was that it would concern 6% of women whereas the true rate of preeclampsia in nulliparous women is closer to 2%. We have added this in the discussion section “Our hypothesis and sample size were built on numbers of outcomes which were not specific from the French population. A French study conducted when the trial was designed found that preeclampsia affected only 1.7% of nulliparous women whereas we used an expected rate of 6% which is erroneous (10). It is likely that the lower than expected rate for the main outcome may be explained, in part, by this erroneous assessment of the rate of preeclampsia in nulliparous women in France(9).”

In the conclusion section, you must add that your results are uncertain because of lack of power related to insufficient number of patients recruited in the study.

We have added this in the conclusion section

Reviewer #3: This is a randomized, blinded, placebo-controlled, parallel-group trial assessing whether daily low-dose aspirin initiated before 16 weeks of gestation can reduce preeclampsia and growth restriction in nulliparous women identified by first-trimester uterine artery Dopplers as at high risk of preeclampsia. Aspirin was administered at the appropriated doe of 160 mg/day. Preeclampsia or a birthweight ≤ 5th percentile occurred in 88 (16.0%) women in the low-dose aspirin group and in 79 (14.4%) in the placebo group (proportion difference 1.6 [-2.6 ; 5.9] p=0.45). The two groups did not differ significantly for the secondary outcomes.

The manuscript by Diguisto et al. is well written. The methodology is clear and clearly explained. Although, the trial was interrupted after including 44% of the expected number of patients (1104 of the 2486 planned), the results of this study are interesting and need to be published.

In the introduction, please mention the prevalence of preeclampsia (PE) in France in the general population and in nulliparous women. 

This will allow to show that the patients of the placebo group are at higher risk of PE.

We have added two references suggested by the reviewer: Subtil et al BJOG 2003 and Goffinet et al BJOG 2001

The result section is very limited. Please describe the results in this section. Gestational age at delivery should be added.

We have added this in the result section in text. This is not a baseline characteristic or an outcome so it did not fit in any of our tables.

In the discussion, please present the results of the previous studies evaluating first trimester uterine Doppler (such The Great Obstetrical Syndrome Study) and show how the outcomes of the placebo group are comparable to these studies. Please also discuss the previous studies assessing the effect of low dose aspirin in patients first trimester uterine Doppler and show how the outcomes of the placebo (Acetylsalicylic acid for the prevention of preeclampsia and intra-uterine growth restriction in women with abnormal uterine artery Doppler: a systematic review and meta-analysis; Usefulness of aspirin therapy in high-risk pregnant women with abnormal uterine artery Doppler ultrasound at 14-16 weeks pregnancy: randomized controlled clinical trial).

Please also discuss that 2/3 of preeclampsia cases occurred after 37WG. This is normal but may contribute to the results of the trial.

We have changed the discussion section and mentioned these two references. We also discuss our findings according to those two studies.

We have added in the discussion section that 2/3 of preeclampsia occurred after 37 WG in our study. Those late/term preeclampsie are not particularly prevented by low dose aspirin which could explain why our trial is negative. This paragraph has been added to the discussion: “Bujold et al’s meta-analysis which evaluated acetylsalicylic acid for the prevention of preeclampsia and intra-uterine growth restriction in women with abnormal uterine artery doppler found that this treatment, when initiated early, could reduce the incidence of preeclampsia and in particular severe preeclampsia. It turns out that most of the observed cases of term preeclampsia and was mild which may explain partly the lack effect of the treatment in our trial.”

Line 121 : the sentence is not clear. Did the patients receive Kardegic sachets or a reconditioned for of aspirin? If patients received Kardegic sachets please explain how the patients and the investigators were blinded to the treatment?

We have changed the control paragraph to make things clearer « Women randomized to the control group received placebo powder (purchased from Bertin Pharma), in similar-looking sachets as the intervention group, to be stored and taken according to the same protocol.“ 

Lines 301-303: The sentence is not clear. Please who was more successful in identifying women at risk.

We have changed this phrase to make it more easier to understand, it is now line 286 “The observed rate of preterm preeclampsia in our control group was half that in the ASPRE trial control group, which means that their algorithm was more successful in identifying women at risk of preeclampsia.” 

6. PLOS authors have the option to publish the peer review history of their article (what does this mean?). If published, this will include your full peer review and any attached files.

Do you want your identity to be public for this peer review? For information about this choice, including consent withdrawal, please see our Privacy Policy.

Reviewer #1: No

Reviewer #2: No

Reviewer #3: No

---

## [Editor Report · Decision Letter 1]

1 Aug 2022

PONE-D-22-07025R1Low-dose aspirin to prevent preeclampsia and growth restriction in nulliparous women identified by uterine artery Doppler as at high risk of preeclampsia: a double blinded randomized placebo-controlled trialPLOS ONE

Dear Dr. Diguisto,

Thank you for submitting your manuscript to PLOS ONE. After careful consideration, we feel that it has merit but does not fully meet PLOS ONE’s publication criteria as it currently stands. Therefore, we invite you to submit a revised version of the manuscript that addresses the points raised during the review process.

ACADEMIC EDITOR:

You made appropriate changes following the Reviewers comments. However, an edit shown in the response to Reviewer #3 was overlooked and not inserted into the edited text: "We have added in the discussion section that 2/3 of preeclampsia occurred after 37 WG in our study. Those late/term preeclampsie are not particularly prevented by low dose aspirin which could explain why our trial is negative. This paragraph has been added to the discussion: “Bujold et al’s meta-analysis which evaluated acetylsalicylic acid for the prevention of preeclampsia and intra-uterine growth restriction in women with abnormal uterine artery doppler found that this treatment, when initiated early, could reduce the incidence of preeclampsia and in particular severe preeclampsia. It turns out that most of the observed cases of term preeclampsia and was mild which may explain partly the lack effect of the treatment in our trial.” Please correct this oversight and modify the text accordingly before final acceptance.

We look forward to receiving your revised manuscript.

Kind regards,

Patrick Rozenberg, MD

Academic Editor

PLOS ONE

Journal Requirements:

Additional Editor Comments:

The authors made appropriate changes following the Reviewers comments.

However, an edit shown in the response to Reviewer #3 was overlooked and not inserted into the edited text:

"We have added in the discussion section that 2/3 of preeclampsia occurred after 37 WG in our study. Those late/term preeclampsie are not particularly prevented by low dose aspirin which could explain why our trial is negative. This paragraph has been added to the discussion: “Bujold et al’s meta-analysis which evaluated acetylsalicylic acid for the prevention of preeclampsia and intra-uterine growth restriction in women with abnormal uterine artery doppler found that this treatment, when initiated early, could reduce the incidence of preeclampsia and in particular severe preeclampsia. It turns out that most of the observed cases of term preeclampsia and was mild which may explain partly the lack effect of the treatment in our trial.”

The authors should correct this oversight and modify the text accordingly before final acceptance.
---

## [Author Response · Author response to Decision Letter 1]

2 Sep 2022

The Editor-in-Chief, Plosone

 September, 2022

Dear Editor,

Please find attached our revised manuscript PONE-D-22-07025-R2 “Low-dose aspirin to prevent preeclampsia and growth restriction in nulliparous women identified by uterine artery Doppler as at high risk of preeclampsia: a double blinded randomized placebo-controlled trial “which we hope is now suitable for publication in Plosone. 

We have made a response to the editor’s comment below.

The authors have declared that no competing interests exist.

Caroline Diguisto, MD, PhD, on behalf of all authors

Pôle de gynécologie obstétrique, médecine fœtale, médecine et biologie de la reproduction, centre Olympe de Gouges, CHRU de Tours, 37 044 Tours, France Tel: +33 2 47 47 47 47 

Email carolinediguisto@gmail.com

EDITOR

You made appropriate changes following the Reviewers comments. However, an edit shown in the response to Reviewer #3 was overlooked and not inserted into the edited text: "We have added in the discussion section that 2/3 of preeclampsia occurred after 37 WG in our study. Those late/term preeclampsie are not particularly prevented by low dose aspirin which could explain why our trial is negative. This paragraph has been added to the discussion: “Bujold et al’s meta-analysis which evaluated acetylsalicylic acid for the prevention of preeclampsia and intra-uterine growth restriction in women with abnormal uterine artery doppler found that this treatment, when initiated early, could reduce the incidence of preeclampsia and in particular severe preeclampsia. It turns out that most of the observed cases of term preeclampsia and was mild which may explain partly the lack effect of the treatment in our trial.” Please correct this oversight and modify the text accordingly before final acceptance.

We have inserted the missing paragraph in the discussion section and apologize for this oversight.

---

## [Editor Report · Decision Letter 2]

12 Sep 2022

Low-dose aspirin to prevent preeclampsia and growth restriction in nulliparous women identified by uterine artery Doppler as at high risk of preeclampsia: a double blinded randomized placebo-controlled trial

PONE-D-22-07025R2

Dear Dr. Diguisto,

We’re pleased to inform you that your manuscript has been judged scientifically suitable for publication and will be formally accepted for publication once it meets all outstanding technical requirements.

Kind regards,

Patrick Rozenberg, MD

Academic Editor

PLOS ONE

Additional Editor Comments (optional):

This manuscript have been appropriately corrected and can be published
---

## [Editor Report · Acceptance letter]

22 Sep 2022

PONE-D-22-07025R2 

Low-dose aspirin to prevent preeclampsia and growth restriction in nulliparous women identified by uterine artery Doppler as at high risk of preeclampsia: a double blinded randomized placebo-controlled trial 

Dear Dr. Diguisto:

I'm pleased to inform you that your manuscript has been deemed suitable for publication in PLOS ONE. Congratulations! Your manuscript is now with our production department. 

Kind regards, 

on behalf of

Professor Patrick Rozenberg 

Academic Editor

PLOS ONE